# Ketogenic Diet in Steatotic Liver Disease: A Metabolic Approach to Hepatic Health

**DOI:** 10.3390/nu17071269

**Published:** 2025-04-04

**Authors:** Fabrizio Emanuele, Mattia Biondo, Laura Tomasello, Giorgio Arnaldi, Valentina Guarnotta

**Affiliations:** 1Department of Health Promotion, Mother and Child Care, Internal Medicine and Medical Specialties “G. D’Alessandro” (PROMISE), Section of Endocrinology, University of Palermo, Piazza delle Cliniche 2, 90127 Palermo, Italy; fabema.diet@gmail.com (F.E.); laura.tomasello@unipa.it (L.T.); giorgio.arnaldi@unipa.it (G.A.); valentina.guarnotta@unipa.it (V.G.); 2Department of Biological, Chemical and Pharmaceutical Sciences and Technologies (STEBICEF), University of Palermo, Viale delle Scienze Ed. 16, 90128 Palermo, Italy; mattia.biondo@unipa.it

**Keywords:** hepatic steatosis, weight loss, fatty liver, ketogenic diet, metabolic dysfunction-associated steatotic liver disease, NAFLD

## Abstract

Metabolic dysfunction-associated steatotic liver disease (MASLD) is a major cause of chronic liver dysfunction worldwide, characterized by hepatic steatosis that may progress to nonalcoholic steatohepatitis and cirrhosis. Owing to its strong association with metabolic disorders, current management focuses on weight reduction via lifestyle modifications. Recently, the very-low-calorie ketogenic diet (VLCKD) has emerged as a promising intervention due to its potential for rapid weight loss and reduction in liver fat. This review aims to evaluate the clinical evidence regarding the impact of ketogenic diets on hepatic steatosis. We conducted an extensive MEDLINE literature search in databases including PubMed, Scopus, and Web of Science up to December 2024. Studies assessing the effects of ketogenic or low-carbohydrate high-fat diets on liver fat, evaluated by imaging, histology, or biochemical markers, were included. The analysis indicates that ketogenic diets significantly reduce hepatic fat content and improve metabolic parameters, including insulin sensitivity and liver enzyme levels. Evidence further suggests that substituting saturated fats with unsaturated fats or replacing carbohydrates with proteins may enhance these benefits. However, considerable variability exists among studies and long-term data remain limited. Although short-term outcomes are encouraging, potential adverse effects such as dyslipidaemia, gastrointestinal disturbances, and transient ‘keto flu’ symptoms require careful clinical monitoring. Future research should focus on elucidating underlying mechanisms, optimizing dietary composition, and assessing long-term safety to establish ketogenic diets as a robust strategy for managing MASLD.

## 1. Introduction

Metabolic dysfunction-associated steatotic liver disease (MASLD) is one of the most prevalent causes of chronic liver disease worldwide. It encompasses a spectrum of hepatic disorders characterized by hepatic steatosis once secondary causes of fat accumulation, such as excessive alcohol intake, have been excluded. Introduced in 1986, the term NAFLD includes two stages: nonalcoholic fatty liver, characterized by minimal inflammation and damage, and nonalcoholic steatohepatitis (NASH), which involves significant liver inflammation and injury leading to liver fibrosis and cirrhosis [1,2].

The connection between NAFLD and metabolic disorders is so substantial that recently, an update of the terminology to metabolic dysfunction-associated steatotic liver disease (MASLD) was proposed. MASLD was defined as steatotic liver disease (SLD) occurring in conjunction with at least one of the five cardiometabolic criteria for adults [3]. However, the available data remain inadequate to establish whether MASLD is effective in identifying metabolic abnormalities and advanced fibrosis, despite indications from certain studies suggesting this possibility [4].

A recent meta-analysis by Riazi et al. reported an incidence of hepatic steatosis at 46.9 cases per 1000 person-years, with a higher incidence in males, though these findings are limited to Asian populations [5]. Previous studies corroborate similar incidence rates across various Asian countries [6]. The global prevalence of SLD is estimated at 32%, rising from 26% before 2005 to 38% in recent studies. However, these data come from only 17 countries, highlighting the need for better data collection in underrepresented regions like Africa, Oceania, and South America [7]. Another analysis by Le et al. indicated a prevalence of 29.8%, aligning with Riazi’s findings and pointing to a significant rise primarily attributed to obesity, metabolic syndrome, insulin resistance, and dyslipidemia [8]. Liver biopsy remains the gold standard for assessing liver inflammation and damage [9]. Histological analysis is key in diagnosing MASLD, revealing triglyceride deposition in hepatocytes and advanced NASH features [10].

Proton magnetic resonance spectroscopy (^1^H-MRS) is regarded as the most precise method for evaluating hepatic triglyceride content and aligns closely with liver biopsy in assessing and classifying MASLD [11]. While it allows for repeated examinations without safety concerns, its high cost, time-intensive nature, and limited availability in specialized centers worldwide pose significant limitations. An alternative widely used ultrasound-based technique for quantifying hepatic steatosis is the controlled attenuation parameter (CAP) obtained by transient elastography, commonly known as FibroScan [12,13].

The primary approach to managing MASLD remains as weight reduction through lifestyle modifications, including increased physical activity and caloric restriction [14,15]. However, conventional low-calorie (LC) diets and exercise regimens often fail to achieve sufficient weight loss to resolve hepatic steatosis, limiting their effectiveness in reversing liver fat accumulation [16]. Recently, resmetirom has been approved by the FDA combined with diet and physical activity in patients with NASH. However, there are no pharmacological therapies approved for MASLD, highlighting an unmet clinical need for effective treatment strategies [17]. In this context, the very-low-calorie ketogenic diet (VLCKD) has emerged as a promising intervention for MASLD due to its ability to induce rapid weight loss and substantial reductions in hepatic steatosis and visceral adiposity [16,18,19]. The macronutrient composition of the ketogenic diet is typically characterized by a high-fat intake (60–80/90% of total energy), moderate protein (10–30%), and very low carbohydrate content (5–10%, generally not exceeding 50 g per day). By promoting ketosis, VLCKD may improve insulin resistance and mitigate end-organ damage associated with metabolic dysfunction [20,21,22].

Despite its efficacy, concerns have been raised regarding the safety of ketogenic diets, particularly high-fat variants, due to their reported effects on serum cholesterol levels and liver function, which could contribute to the onset or exacerbation of hepatic steatosis [23]. Conversely, the potential benefits of ketogenic diets in MASLD have also been explored, given emerging evidence that ketone bodies may exert anti-inflammatory effects and ameliorate metabolic dysfunction implicated in MASLD pathogenesis [24,25].

This review aims to critically assess and systematically synthesize the latest clinical evidence on the effects of the ketogenic diet in hepatic steatosis, addressing current controversies and identifying future research directions. This critical appraisal will help contextualize the ketogenic diet within the evolving therapeutic landscape of MASLD.

## 2. Materials and Methods

An extensive MEDLINE search was conducted across multiple electronic databases, including the following: PubMed/MEDLINE, Scopus, and Web of Science (until 30 December 2024).

The search strategy was restricted to papers published in English and incorporated a combination of Medical Subject Headings (MeSH) terms and free-text keywords. The primary terms used included: (Ketogenic diet AND fatty liver disease) OR (Ketogenic diet AND NAFLD) OR (Ketogenic diet AND hepatic steatosis) OR (low-carbohydrate high-fat diet AND fatty liver disease) OR (low-carbohydrate high-fat diet AND NAFLD) OR (low-carbohydrate high-fat diet AND hepatic steatosis) OR (ketogenic diet AND MASLD).

Search filters were applied to exclude preclinical studies, case reports, and non-English publications.

## 3. Mechanisms of Action and Rationale of the Ketogenic Diet in Steatotic Liver Disease

The liver is the organ mainly involved in the synthesis of ketone bodies [26,27]. Ketogenesis is promoted by a condition of limited glucose availability, favored by the ketogenic diet, as well as by pathological conditions such as diabetic ketoacidosis.

In conditions of glucose deficiency, with insufficient pyruvate and oxaloacetate for gluconeogenesis, Acetyl-CoA cannot enter the Krebs cycle [28]. This results in an excess of acetyl-CoA and consequently the activation of the metabolic pathway of ketogenesis, which leads to the production of ketone bodies: acetoacetate, beta-hydroxybutyrate, and acetone [29]. This also makes CoA, present in limited amounts and essential for the continuation of fatty acid beta-oxidation processes, continuously available.

Within the ketogenesis pathway, two molecules of acetyl-CoA combine via a condensation reaction to yield acetoacetate, with the concurrent release of two molecules of CoA. Additionally, part of acetoacetate is synthesized in renal tissue through the direct hydrolytic cleavage of acetoacetyl-CoA. As the principal ketone body, acetoacetate may subsequently be reduced to β-hydroxybutyrate in an NADH-dependent reaction or undergo decarboxylation to form acetone. Ketogenesis is intricately regulated at several levels. This regulation encompasses the control of free fatty acid synthesis, the hepatic routing of these fatty acids towards either esterification or β-oxidation, and the allocation of acetyl-CoA between oxidation in the Krebs cycle and condensation for ketone body synthesis [30].

Ketone bodies produced by hepatocytes are utilized by muscle, neural cells, the heart, and the brain for oxidative metabolism to generate energy. In the kidney, these compounds are partially oxidized and partially excreted in the urine (ketonuria). When glucose availability is limited or absent, hepatic ketogenesis becomes indispensable for neural cells, which lack the capacity for fatty acid β-oxidation. In contrast, in the heart and muscle tissue, liver-derived ketone bodies serve as an alternative substrate that enhances fatty acid oxidation. Consequently, the metabolic interplay between the liver and neural tissue during states of glucose deficiency is of critical importance.

In healthy individuals with sufficient carbohydrate intake, plasma ketone body concentrations typically range from 0.3 to 2 mg per 100 mL. When the liver produces ketones at a rate that exceeds the capacity of peripheral tissues to utilize them, these compounds accumulate in the bloodstream, leading to ketosis, and are excreted in the urine as ketonuria. Due to their acidic properties, prolonged ketosis may progress to ketoacidosis, a severe and potentially life threatening complication in cases of insulin deficient diabetes [31,32]. However, ketoacidosis develops only in situations of severe glucose deficiency or complete insulin deficiency, as the body employs homeostatic mechanisms to regulate pH. For example, enhanced production of ammonium ions (NH4+) and bicarbonate helps prevent progression to acidosis.

We can define the state of ketosis induced by the ketogenic diet as a ‘controlled’, physiological ketosis in which the pH is not altered [33]. Furthermore, in healthy individuals, a rise in circulating ketone bodies stimulates insulin secretion. This increased insulin not only suppresses hepatic ketogenesis and the mobilization of free fatty acids from adipose tissue, but also enhances the urinary excretion of ketone bodies (Figure 1).

A potential pathophysiological basis for employing the ketogenic diet in managing MASLD may relate to the influence of ketogenesis on insulin resistance. Insulin resistance drives de novo lipogenesis, suppresses the oxidation of FFAs and accelerates the breakdown of very low-density lipoproteins (VLDL) in the liver. These mechanisms collectively promote excessive fat accumulation within the liver.

Given the well-documented relationship between insulin resistance and hepatic steatosis, the long-term mortality risks associated with fatty liver disease may arise, in part, from its capacity to perpetuate insulin resistance. Over time, this cycle may foster the development and progression of diabetes mellitus and its related complications, such as cardiovascular disease and chronic kidney damage.

A hallmark of insulin-resistant conditions, including obesity, hepatic steatosis and type 2 diabetes, is the persistent and excessive reliance on FFA as the primary energy source for the liver and muscles, at the expense of glucose utilization. This chronic oversupply of FFA, characteristic of insulin resistance, is referred to as lipotoxicity [34].

Beyond its well-established effect on body weight, the ketogenic diet has demonstrated significant benefits in improving insulin resistance. Notably, these effects on insulin resistance are, at least in part, independent of weight loss, thereby expanding the potential applications of this dietary approach [35,36].

Hepatic glucose production is influenced partly by hepatic insulin sensitivity, which is impaired by the excessive accumulation of fatty acids in hepatocytes. In recent years, a growing body of evidence has shown that patients with hepatic steatosis undergoing a ketogenic diet experience a rapid and substantial reduction in hepatic fat, accompanied by significant weight loss [37,38]. The reduction in hepatic triglycerides improves hepatic insulin sensitivity by decreasing excessive hepatic glucose production and compensatory hyperinsulinaemia. Furthermore, lower glucose and insulin levels associated with ketogenic diets downregulate cholesterol biosynthesis via reduced activation of β-hydroxy β-methylglutaryl-CoA reductase by insulin [39].

These effects highlight the potential therapeutic role of the ketogenic diet in patients with hepatic steatosis, even in the absence of weight loss, and support its efficacy in this clinical setting.

However, the EASL-EASD-EASO clinical practice guidelines state that the role of the ketogenic diet in the prevention and treatment of MASLD is not supported by significant evidence, although it does not exclude a positive impact on liver health. Currently, there are no specific recommendations for a ketogenic diet approach to managing MASLD, unlike a Mediterranean diet approach and physical activity. Considering that in adults with MASLD and overweight, weight loss induced by dietary and behavioral therapy could improve hepatic steatosis (with at least a 5% weight reduction), we can assume that the ketogenic diet could help to achieve this goal in this patient category [40].

Moreover, recently, Cai et al. reported in high-fat diet-fed mice, a decrease in circulating bile acids pool, impairing lipid homeostasis and gut microbiota, and resulting in the obese phenotype and metabolic disorders. This evidence showed the potential risks of a long-term high-fat diet and highlighted the relationship among gut microbiota, lipid digestion and bile acid metabolism, which could be positively influenced by the ketogenic diet in patients with SLD [41].

## 4. Clinical Studies

The literature occasionally presents conflicting data regarding the ketogenic diet’s impact on metabolic profile and liver function. A key consideration is that, regardless of body weight, obesity-related morbidity and mortality depend on the distribution of adipose tissue, particularly in tissues that rely on insulin and within hepatocytes [42]. This pattern of fat distribution seems to influence the development of SLD and contributes to insulin resistance. However, it remains uncertain whether these metabolic alterations are a cause or a consequence of SLD [43]. Moreover, evidence shows that the ketogenic diet leads to greater weight loss compared to other low-calorie approaches and enables a more rapid attainment of weight loss targets [44]. In addition, it can reduce liver fat and lower the risk of developing MASLD (Table 1).

One of the first studies investigating the role of the ketogenic diet was performed by Colles et al. who included 32 subjects (19 men and 13 women) with severe obesity for 12 weeks. Dietary compliance was assessed by using urinary ketone reagent strips. A significant decrease in body weight, liver volume, and visceral/subcutaneous adipose tissue, evaluated by computed tomography, were observed in the early phase (between weeks 0 and 2). Additionally, a significant decrease in alkaline phosphatase (ALP), glutamic oxaloacetic transaminase (GOT), glutamic piruvate transaminase (GPT), γ-glutamyltransferase (GGΤ) and bilirubin was also reported [45].

Tendler et al. recruited five obese patients with histologically confirmed hepatic steatosis and subjected them to a calorie-unrestricted high-fat ketogenic diet (HFKD) for six months. In addition to an overall weight loss of 10.9%, four patients showed improvements in steatosis, necro-inflammatory grade, and fibrosis [46]. The subjects adhered to a low-carbohydrate ketogenic diet (LCKD) with carbohydrate intake limited to less than 20 g per day; comparisons of pre- and post-intervention liver biopsies revealed significant improvements, suggesting a potential reversal of cirrhotic changes. In this study dietary compliance was assessed by measurement of ketone levels in urine samples collected from each patient during group meetings. It should be noted that the absence of a comparative treatment group limits the ability to attribute these effects solely to the LCKD, as improvements may also have been influenced by enhancements in fasting insulin and glucose levels.

In a Spanish study evaluated the Spanish Ketogenic Mediterranean Diet (SKMD) in 14 overweight male subjects with metabolic syndrome and SLD. Significant improvements in transaminase levels, steatosis severity, and various metabolic parameters were observed after 12 weeks of SKMD protocol. A regression of metabolic syndrome occurring more rapidly than both weight loss and the resolution of hepatic steatosis was also observed [47]. Patients measured their body’s ketosis state by ketone urine-testing strips every day, and the status was confirmed every week by the physician using ketone blood testing strips.

Similarly, Cunha et al. [38] demonstrated a greater reduction in liver fat and proton density fat fraction (PDFF) by nuclear magnetic resonance imaging in 22 patients treated with a VLCKD compared with 24 patients on a low-calorie diet (LCD) after 2 months of diet. Both groups underwent periodic clinical assessments by a specialized endocrinologist through the course of the study. This study investigated liver morphological changes during weight loss in obese patients, supporting a potential therapeutic role for VLCKD in MASLD.

In a separate investigation on obese and overweight individuals aged 18–64 years who were not receiving drug therapy, 8 weeks of VLCKD produced general clinical improvements, including increased vitamin D levels, improved CAP and stiffness on FibroScan, meaning decreased steatosis, fibrosis, and inflammation and reductions in white blood cell and platelet counts [54]. Additionally, a study comparing the hepatic response following a ketogenic diet, with and without ketone supplementation, to that following a hypolipidic diet found that weight loss effects were superimposable, confirming that increased lipid consumption within a reduced-carbohydrate framework did not negatively affect the liver in the short term [53]. Liver fat, assessed by MRI, significantly decreased after six weeks with no differences between groups. In participants with SLD, the reduction was more pronounced but remained similar across groups. Both hypocaloric low-fat diets and KDs effectively reduced liver fat in SLD over the short term. No significant differences were found in hepatic function (serum parameters).

Additional studies have confirmed the benefits of ketogenic diets in hepatic steatosis. A recent real-world study by Rinaldi evaluated a VLCKD in 33 overweight and obese subjects over 8 weeks, demonstrating significant reductions in BMI, waist circumference, fat mass, and fatty liver accumulation (assessed by FibroScan), alongside improvements in fasting blood glucose, insulin, lipid profile, liver enzymes, and blood pressure [55].

Watanabe et al. evaluated 65 obese patients treated for 45 days with a VLCKD followed by 45 days of hypocaloric diet, resulting in a significant decrease in visceral adipose tissue, body fat, hepatic steatosis index (HSI), and GPT values. However, the most significant reductions were observed in the second phase of the diet protocol (in the low calorie diet step), notably for HSI values which decreased below the cut off of 36 [51]. Semi-quantitative determination of acetoacetic acid was measured in the first morning urine at baseline and every other week until the end of the study. Interestingly, the SLD improvement was non-linear with weight loss, supporting that the improvement in liver injury is slower than the other metabolic parameters.

Luukkonen et al. demonstrated that a short-term ketogenic diet (lasting six days) reduced intrahepatic triglyceride levels and improved hepatic insulin resistance despite an increase in plasma non-esterified fatty acid concentrations. Compliance was verified by 3-day food records and an increase in fasting blood ketone concentrations. In a cohort of 10 adult patients, these effects were attributed to enhanced net hydrolysis of intrahepatic triglycerides and the diversion of the resulting fatty acids towards ketogenesis, processes driven by reductions in serum insulin and hepatic citrate synthase flux [52]. In addition, IHTG content decreased by ~31%, while liver stiffness as determined by magnetic resonance remained unchanged. Activities of plasma GGΤ and ALP significantly decreased, while serum GOT and GPT remained unchanged during the diet.

Sila et al. showed the effects of 8 weeks’ VLCKD in 111 overweight/obese patients showing a significant decrease in liver controlled attenuation parameter (CAP), GPT, γ-GT, fat mass, and lipids [57]. No information about the evaluation of the ketosis state is available.

Recently, a randomized controlled study evaluated 24 patients with MASLD randomly assigned to either a ketogenic diet provided through home delivery, or a nutritional education program aimed at promoting adherence to the DASH diet for 8 weeks. No significant change was observed in liver stiffness and steatosis grade, while a significant in fat mass, visceral fat, and GOT levels was shown in patients assigned to ketogenic diet [48]. Serum ketone levels were monitored to confirm the ketosis state.

### 4.1. Studies Conducted on Patients Scheduled for Bariatric Surgery

Preoperative interventions have also been explored. Leonetti et al. [48] evaluated 50 obese patients scheduled for bariatric surgery who followed a VLCKD for 10 days, followed by a LCKD for a further 10 days, and then a low-calorie diet for 10 days. The diet resulted in an improvement of the steatosis pattern and a mean 30% reduction in liver volume, evaluated by ultrasonography. No changes in hepatic serum parameters were found. Similarly, Schiavo et al. conducted a prospective study involving 27 morbidly obese patients scheduled for bariatric surgery, who typically present with steatosis livers and micronutrient deficiencies. Participants followed a four-week preoperative ketogenic micronutrient-enriched diet (KMED) consisting of 4% carbohydrates, 71% fats, and 25% proteins, supplemented with specific nutritional additives. This regimen was safe and resulted in reductions in body weight and left hepatic lobe volume (mean reduction of 19.8%), correction of micronutrient deficiencies (including vitamin B12, folic acid, iron, and zinc), and improvements in liver enzymes, glycaemic control, and lipid profiles [49].

Additionally, Ministrini et al. evaluated 52 obese patients scheduled for bariatric surgery undergoing a 25-day VLCKD. All patients had a positive test for urinary ketones up to 72 h after the beginning of the diet. A significant decrease in GOT, GPT, GGT, and liver steatosis grade, evaluated by ultrasonography, were found. In addition, the low activity of lysosomal acid lipase (LAL) activity was measured in 20 obese patients and in a control group of 20 healthy, normal-weight subjects, with a significant increase after diet in obese patients. LAL activity was found to be related with BMI and weight, suggesting a role in promoting recovery from SLD and enhance lipolysis [50].

### 4.2. Studies on Gender Differences

In 2024, Rinaldi and colleagues further examined the impact of gender on hepatic outcomes in 112 overweight or obese patients with MASLD undergoing an eight-week VLCKD. Although both sexes experienced weight loss and improved lipid profiles, men exhibited higher levels of steatosis and fibrosis compared with women, with steatosis remaining elevated in men even after the intervention. The authors suggested that a gender-specific clinical approach may be appropriate [56]. In this light, it has been demonstrated that women with MASLD in premenopausal age are protected from hepatic fibrosis and hepatocellular carcinoma, due to the protective role of estrogens from visceral obesity, insulin-resistance, and MASLD [59,60]. In addition, men consume more alcohol than women and tend to have an unhealthy diet eating more meat and fatty and junk food than women who consume more fruit and vegetables [61,62].

### 4.3. Clinical Studies with Technological Enhancements

Technological innovations have also been integrated with ketogenic dietary approaches. Falkenhain et al. [63] compared two mobile health applications aimed at weight loss, including 155 participants with overweight or obesity for 12 weeks. One app promoted a Mediterranean-style ketogenic diet with breath acetone biofeedback, while the other focused on a calorie-restricted, low-fat diet. Participants used a wireless scale to record their weight daily, and blood samples were collected before and after 12 weeks of intervention. Those using the ketogenic diet app experienced more substantial weight loss at 12 weeks, and this effect remained evident at 24 weeks. Moreover, improvements in glycemic control and ALP and GPT levels were greater in the ketogenic group. Overall, the findings suggest that the ketogenic diet app with breath acetone biofeedback is more effective in promoting weight loss in a real-world setting than the calorie-restricted, low-fat diet app.

## 5. Adverse Events

The ketogenic diet, characterized by a high-fat and low-carbohydrate intake, is associated with a range of side effects, particularly in the initial phase following the introduction of the new dietary regimen [64].

### 5.1. Ketogenic Flu

One of the most commonly reported transient effects during the first few weeks of a ketogenic diet is the so-called ‘ketogenic flu’ or ‘keto flu’, characterized by symptoms such as fatigue, headaches, nausea, and irritability, which result from the body’s adaptation to the new metabolic state [65]. This condition, while not scientifically classified as a formal diagnosis, is frequently reported by individuals following a ketogenic diet during the initial phase. The symptoms generally subside as the body adapts, leading to improved tolerance to the dietary changes. From a pathophysiological perspective, this phenomenon arises due to the body’s transition from glucose-based metabolism to ketone body utilization as the primary energy source.

### 5.2. Lipid Metabolism

Regarding the potential adverse metabolic effects of the ketogenic diet, one of the primary concerns is its impact on lipid metabolism. The ketogenic diet has been associated with an increase in low-density lipoprotein (LDL) cholesterol levels, primarily due to its inherently high fat content. This concern has led many clinicians to exercise caution in recommending this dietary approach, particularly due to the potential risk of exacerbating or inducing dyslipidaemia, especially in the long term [66]. Some studies have suggested that adherence to a ketogenic diet may elevate the risk of cardiovascular disease, particularly if unhealthy dietary choices are made, such as the excessive consumption of saturated fats, while neglecting sources of unsaturated fats like olive oil and avocado. However, the effects on lipid metabolism are highly individualized, as metabolic responses to increased dietary fat intake vary significantly among individuals. Consequently, it is recommended to monitor the lipid profile both before initiating the ketogenic diet and periodically throughout its course. Indeed, cases have been reported in which individuals with previously normal lipid profiles developed severe dyslipidaemia following the introduction of the ketogenic diet [67].

In individuals who exhibit a pronounced increase in LDL cholesterol levels, often referred to as ‘hyper-LDL responders’, the underlying pathophysiological mechanism appears to involve enhanced intestinal cholesterol absorption. In such cases, the administration of ezetimibe has been reported to aid in lipid balance regulation. Due to the potential for worsening lipid profiles, the ketogenic diet is generally contraindicated in individuals with pre-existing dyslipidaemia [68]. Furthermore, given the lack of long-term studies on its cardiovascular effects, current recommendations suggest that the ketogenic diet should be used primarily for its effects on body weight, for a limited duration, and generally for no longer than one year [69]. Regardless of an individual’s baseline lipid status, periodic biochemical monitoring of lipid parameters is advised throughout the duration of the diet [70]. Additionally, further research is warranted to investigate the role of specific genetic variants in modulating individual responses to lipid metabolism alterations induced by the ketogenic diet.

### 5.3. Gastrointestinal Side Effects and Electrolyte Balance

Another commonly reported adverse effect of the ketogenic diet pertains to gastrointestinal function. The reduction in carbohydrate intake, coupled with increased fat consumption, can lead to changes in normal bowel transit. Many individuals experience constipation due to the reduced intake of fiber-rich foods such as fruits, vegetables, and whole grains, which are significantly restricted in the ketogenic diet. Crabtree et al. [53] observed increased levels of aminotransferases in response to a therapeutic KD in children with epilepsy, even if this effect may be attributed to valproate medication, which is commonly prescribed in this population. Additionally, the increased fat intake may contribute to gastrointestinal discomfort, including bloating, nausea, and diarrhea, particularly in the early stages of dietary adaptation. Luukkonen et al. [52] reported that during KD, the AST/ALT ratio increased by ~34%, suggesting that such a rapid weight loss could induce a transient hepatocellular damage.

Hydration and electrolyte balance are also critical considerations for individuals following a ketogenic diet. Many individuals experience varying degrees of dehydration and electrolyte imbalances, which may manifest as symptoms such as headaches, muscle cramps, fatigue, and palpitations. It is therefore essential that individuals adhering to a ketogenic diet ensure adequate intake of sodium, potassium, and magnesium, maintain sufficient hydration, and incorporate foods rich in these essential minerals. Indeed, documented electrolyte disturbances associated with the ketogenic diet include hyponatraemia, hypomagnesaemia, and hyperuricaemia [71].

### 5.4. Nephrolithiasis and Bone Health

A further potential risk of the ketogenic diet is an increased predisposition to nephrolithiasis (kidney stones). This is attributed to the heightened urinary acidity induced by ketone body excretion and the restricted carbohydrate intake, which can disrupt mineral homeostasis. The estimated incidence of kidney stones in individuals on a ketogenic diet is approximately 5.9% (5.8% in children and 7.9% in adults). The majority of these cases involve uric acid stones, followed by calcium-based stones [72]. Although the overall prevalence is relatively low, individuals with a predisposition to renal complications should approach the ketogenic diet with caution and under medical supervision.

Regarding the long-term effects of the ketogenic diet, its precise impact remains difficult to ascertain due to the general recommendation that it be followed for a limited period. Most researchers consider the ketogenic diet an unsustainable long-term dietary approach. One of the key metabolic concerns requiring further investigation is its potential effect on bone health. At present, no human studies have specifically examined the long-term impact of the ketogenic diet on osteoporosis. However, short-term assessments of bone health parameters following ketogenic diet interventions have not demonstrated significant detrimental effects on bone integrity [73].

## 6. Discussion

The ketogenic diet has emerged as a promising therapeutic approach for managing metabolic dysfunction, particularly due to its ability to reduce hepatic steatosis and improve markers of insulin resistance and adiposity. Growing evidence supports its short-term efficacy in promoting rapid weight loss and metabolic recalibration. Several studies have demonstrated a significant impact on improving MASLD, assessed both through imaging techniques and specific hepatic steatosis indices.

However, studies investigating the effects of the ketogenic diet in MASLD exhibit substantial heterogeneity. This variability arises from differences in sample size, ranging from as few as 5 to as many as 112 patients, as well as in the duration of dietary intervention, which varies between 6 days and 6 months. Additionally, the methodologies used to assess hepatic steatosis differ across studies.

Earlier studies primarily relied on radiological assessments, such as ultrasound, which is operator-dependent, or computed tomography and magnetic resonance imaging, which offer greater sensitivity in detecting liver fat changes. More recently, hepatic steatosis has been quantified using CAP, a specific technology coupled with FibroScan. Some studies included in this review also assessed hepatic steatosis using surrogate biomarkers, including HSI and FLI, all of which demonstrated improvement following the ketogenic diet.

In addition, fibrosis, the hallmark of disease progression, has been evaluated by surrogate biomarkers, FIB-4, and by liver stiffness at FibroScan, showing an improvement in the studies included in this review.

Moreover, all studies reported a significant reduction in GGT levels, while most also documented a decrease in GOT and GPT levels.

Notably, the duration of the dietary protocol plays a crucial role in the magnitude of these effects, with longer interventions yielding more pronounced metabolic and hepatic benefits.

Contrary to the assumption that high dietary fat intake exacerbates hepatic steatosis, evidence suggests that a normocaloric high-fat KD can suppress de novo lipogenesis and enhance fatty acid oxidation, thereby benefiting liver health independently of total fat intake [74,75]. Meta-analyses indicate that replacing carbohydrates with proteins or unsaturated fats may further improve hepatic outcomes, whereas substituting saturated fat with unsaturated fat appears to mitigate liver fat accumulation [76].

In addition, sex-specific responses to KD have also been reported, with estrogen-mediated differences influencing lipid metabolism. Women, particularly premenopausal individuals, may experience altered ketosis efficiency due to hormonal fluctuations, necessitating tailored dietary strategies. Socio-cultural factors further contribute to adherence variability, underscoring the need for personalized interventions. Concurrently, socio-cultural factors, such as gender roles in food preparation or dietary adherence and alcohol consumption further show an impact in MASLD outcomes during KD.

Despite the effectiveness in improving MASLD, concerns remain regarding KD safety. While transient effects such as “keto flu” and gastrointestinal discomfort are common, potential risks include dyslipidaemia, micronutrient deficiencies, and renal strain are rare, and many studies showed also an improvement in lipid profile. These challenges highlight the need for careful monitoring, with mitigation strategies including electrolyte supplementation, prioritization of unsaturated fats, and periodic metabolic assessments.

Key knowledge gaps persist, particularly regarding the diet’s long-term effects on cardiovascular health and hepatic steatosis. Future research should integrate precision nutrition frameworks, leveraging biomarkers to optimize patient selection while taking into account inter-individual heterogeneity driven by genetic predisposition, gut microbiota composition, lifestyle factors, and adherence variability, all of which profoundly influence therapeutic outcomes [77].

In addition, hybrid interventions combining cyclical ketosis with Mediterranean or low-glycaemic principles may enhance sustainability and attenuate risks.

## 7. Conclusions

The ketogenic diet shows considerable promise in improving metabolic dysfunction and reducing hepatic fat, but its clinical use must be approached with caution and personalized care. The variability in individual responses, the presence of gender-specific outcomes, and the range of potential adverse effects emphasize the need for thorough metabolic monitoring and customized intervention strategies. Future research should focus on elucidating the underlying mechanisms, refining patient selection, and assessing the long-term safety and efficacy of ketogenic interventions across diverse populations.

## Figures and Tables

**Figure 1 nutrients-17-01269-f001:**
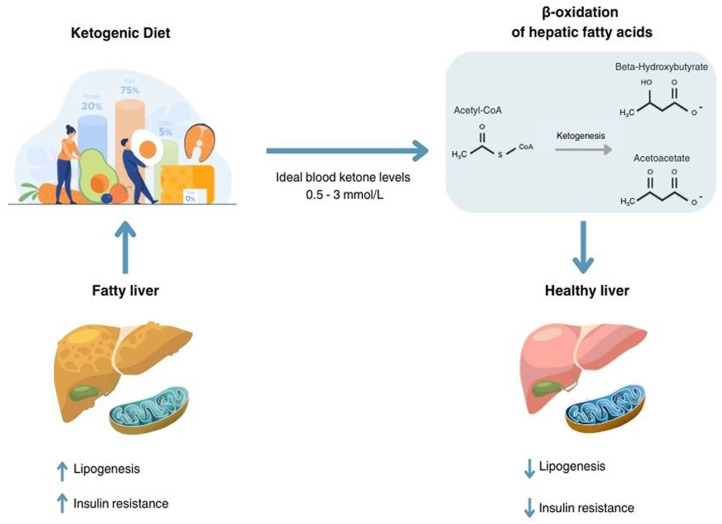
Effects of ketogenic diet in fatty liver disease.

**Table 1 nutrients-17-01269-t001:** Clinical studies focusing on the effects of ketogenic diet in steatotic liver disease.

Author, Year	Type of Study	Diet Protocol	N° Patients Included	Inclusion Criteria	Duration of the Study	Liver Outcomes
Colles et al., 2006 [45]	Prospective, observational	VLCKD	32	Age 18–60 years BMI ≥ 50 kg/m^2^ (women) BMI ≥ 40 kg/m^2^ (men)	12 weeks	Improvement in ALP, GPT, GGT, bilirubin, mean visceral adipose tissue.
Tendler et al., 2007 [46]	Prospective, single arm	LCKD	5	Age 18–65 years BMI ≥ 30 kg/m^2^ (but not >40 kg/m^2^) aminotransferase levels (>1.5 ULN in at least 2 separate occasions)	6 months	Improvement in liver histology
Perez-Guisado et al., 2011 [47]	Prospective	Spanish Ketogenic Mediterranean Diet	14	Transaminase levels > 3ULN Plasma creatinine ≤ 1.3 mg/dL and urea ≤ 40 mg/dL No previous gout or high uric acid No alcoholic, and smoking habits, pregnancy or lactating, BMI ≥ 30 kg/m^2^ Age 18–65 years	12 weeks	Improvement in GOT, GPT, steatosis degree by ultrasonography
Leonetti et al., 2015 [48]	Prospective	VLCKD for 10 days LCKD for 10 days LCD for 10 days	48	Age 18–67 years BMI > 40 kg/m^2^ ASA status I to III Pre bariatric surgery	1 month	Improvement of the steatosis pattern and a mean 30% reduction in liver volume, evaluated by ultrasonography
Schiavo et al., 2018 [49]	Prospective	LCKD	27	Obese patients scheduled for bariatric surgery	1 month	Improvement in GOT, GPT, GGT, left hepatic lobe volume evaluated by ultrasonography
Ministrini et al., 2019 [50]	Prospective	VLCKD	52	Patients scheduled for bariatric surgery Age 18–65 years BMI > 40 kg/m^2^ BMI > 35 kg/m^2^ and obesity related comorbidities	25 days	Improvement in GOT, GPT, GGT, visceral fat and liver steatosis grade
Cunha et al., 2020 [38]	Randomized, prospective	VLCKD LCD	24 VLCKD 24 LCD	Age > 18 years BMI > 30 kg/m^2^	2 months	Improvement in MR liver fat fraction, liver stiffness
Watanabe et al., 2020 [51]	Observational, prospective	VLCKD for 45 days followed by LCD for a further 45 days	45	Age 18–60 years BMI > 30 kg/m^2^ HIS > 36	3 months	Improvement in visceral adipose tissue mass, GOT, GPT, HSI
Luukkonen et al., 2020 [52]	Prospective	LCKD	10	Age 18–70 years Alcohol consumption < 20 g/d for women and <30 g/d for men	6 days	Improvement in ALP, GGT, insulin sensitivity and intrahepatic triglycerides content
Crabtree et al., 2021 [53]	Randomized, prospective	KD + ketone supplementation KD + placebo	28	Age 21–65 years BMI 25–34.9 kg/m^2^	6 weeks	Improvement in liver fat, ALP, HSI
De Nucci et al., 2023 [54]	Real-life prospective	VLCKD	87	Age 18–64 years BMI > 25 kg/m^2^	8 weeks	Improvement in GOT, GPT, GGT, liver fat content by FibroScan, liver stiffness
Rinaldi et al., 2023 [55]	Real-life prospective	VLCKD	33	Age > 18 years BMI > 25 kg/m^2^	8 weeks	Improvement in FLI, hepatic fat by FibroScan, liver stiffness by FibroScan, GPT, GGT
Rinaldi et al., 2024 [56]	Real-life prospective	VLCKD	112	Age 18–65 BMI > 25 kg/m^2^	8 weeks	Improvement in liver steatosis and stiffness by FibroScan
Sila et al., 2024 [57]	Real-life prospective	VLCKD	111	Age 18–64 BMI > 25 kg/m^2^	8 weeks	Decrease in GGT, GPT, controlled attenuation parameter, lipids and fat mass
Chirapongsathorn et al., 2025 [58]	Randomized controlled study	12 ketogenic diet 12 DASH diet	24	MASLD diagnosis by ultrasound, CT, or MRI OR FibroScan (CAP > 200 dB/m)	8 weeks	Decrease in AST, triglyceride, and HDL

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
