# Peer review of "Ketogenic Diet in Steatotic Liver Disease: A Metabolic Approach to Hepatic Health"

_nutrients, 2025, doi:10.3390/nu17071269_

Round 1
Reviewer 1 Report
Comments and Suggestions for Authors
This review aims to systematically evaluate the clinical evidence regarding the impact of ketogenic diets on Non-Alcoholic Fatty Liver Disease. There are some suggestions for the study:
- The type of paper is review, it seems like a meta-analysis review according to the methods. However, it doesn’t contain any meta-analysis result. If these papers are just searched for prepare normal review, it doesn’t need to introduce the search strategy. Therefore, the structure of the paper should be constructed in a better way.
- The abstract is not a typical form of review paper, and it usually don’t contain the word “discussion:”.
- The potential benefits of ketogenic diet in NAFLD are not fully discussed in section 3.
- In section 5, subtitle should be added to divided those paragraphs.
- The potential effect of ketogenic diet on gut microbiota, lipid digestion and bile acid metabolism has been studied. These pathways or function are quite related to NAFLD, and can be discussed. A literature is recommended to be cited and support the lipid metabolism regulation based on bile acid alteration in high-fat intake: Cai H ,Zhang J ,Liu C , et al.High-fat diet-induced decreased circulating bile acids contribute to obesity associated with gut microbiota in mice[J].Foods,2024,13(5)
- Key words should contain ketogenic diet, NAFLD
Author Response
- The type of paper is review, it seems like a meta-analysis review according to the methods. However, it doesn’t contain any meta-analysis result. If these papers are just searched for prepare normal review, it doesn’t need to introduce the search strategy. Therefore, the structure of the paper should be constructed in a better way.
Thank you for your suggestion. We apologize for the misunderstanding related to the article category. Our aim was to write a Review article. We have added our search strategy to improve the transparency of the preparatory phase and the validity of what we have presented in the manuscript. However, according to your comments, we have implemented some changes in the paper to improve its structure.
- The abstract is not a typical form of review paper, and it usually don’t contain the word “discussion:”
The abstract section has been modified to make it more adequate to the paper category.
- The potential benefits of ketogenic diet in NAFLD are not fully discussed in section 3.
We thank you for this comment: we have revised section 3 to clarify some pathophysiological processes which can help to understand the multiple potential positive effects of ketogenic diet in NAFLD.
A potential pathophysiological basis for employing the ketogenic diet in managing NAFLD may relate to the influence of ketogenesis on insulin resistance. Insulin resistance drives de novo lipogenesis, suppresses the oxidation of FFAs and accelerates the breakdown of very low-density lipoproteins (VLDL) in the liver. These mechanisms collectively promote excessive fat accumulation within the liver.
Given the well-documented relationship between insulin resistance and hepatic steatosis, the long-term mortality risks associated with fatty liver disease may arise, in part, from its capacity to perpetuate insulin resistance. Over time, this cycle may foster the development and progression of diabetes mellitus and its related complications, such as cardiovascular disease and chronic kidney damage.
A hallmark of insulin-resistant conditions, including obesity, NAFLD and type 2 diabetes, is the persistent and excessive reliance on FFA as the primary energy source for the liver and muscles, at the expense of glucose utilization. This chronic oversupply of FFA, characteristic of insulin resistance, is referred to as lipotoxicity
- In section 5, subtitle should be added to divide those paragraphs.
Thank you for your comment. We added subtitles in section 5.
- The potential effect of ketogenic diet on gut microbiota, lipid digestion and bile acid metabolism has been studied. These pathways or function are quite related to NAFLD, and can be discussed. A literature is recommended to be cited and support the lipid metabolism regulation based on bile acid alteration in high-fat intake: Cai H ,Zhang J ,Liu C , et al.High-fat diet-induced decreased circulating bile acids contribute to obesity associated with gut microbiota in mice[J].Foods,2024,13(5)
We truly appreciate the suggestion of this paper. We included it in the references and briefly discussed its results and the possible relationship with the ketogenic diet in NAFLD at the end of section 3.
Moreover, recently Cai et al. reported in high-fat diet-fed mice a decrease in circulating bile acids pool, impairing lipid homeostasis and gut microbiota, and resulting in obese phenotype and metabolic disorders. This evidence showed the potential risks of a long-term high-fat diet and highlighted the relationship among gut microbiota, lipid digestion and bile acid metabolism, which could be positively influenced by the ketogenic diet in patients with NAFLD.
- Key words should contain ketogenic diet, NAFLD
Thank you for the advice, we added those keywords to the already provided list.
Reviewer 2 Report
Comments and Suggestions for Authors
The study evaluated the potential metabolic approach to liver health by the ketogenic diet in nonalcoholic fatty liver disease. The study presents interesting data, but I have several comments to which responses should improve the overall interest of the manuscript
1- It is not mentioned whether lowering body weight in the same way as following a ketogenic diet would have a higher benefit? a lower benefit? with regard to hepatic steatosis.
2- Please introduce the flowchart leading to the selected studies
3- Were markers of ketosis measured in the selected studies? This would enable to compare the efficacy of the diets in all the selected studies.
4- the paragraph on adverse effects: it is not clear whether the authors have extracted the information from the selected studies... the data from the selected studies could be more cross-referenced.
5- The authors' conclusions are more the sum of the individual findings of the various articles than a complete synthesis of these different studies.
6- Regarding gender disparities, several papers have been published on the bubject by Mauvais-Jarvis. This author should be cited
Author Response
- It is not mentioned whether lowering body weight in the same way as following a ketogenic diet would have a higher benefit? a lower benefit? with regard to hepatic steatosis.
Thank you for your valuable feedback. We have addressed your comment by adding the following section to paragraph 3, lines 157-160.
" Beyond its well-established effect on body weight, the ketogenic diet has demonstrated significant benefits in improving insulin resistance. Notably, these effects on insulin resistance are, at least in part, independent of weight loss, thereby expanding the po-tential applications of this dietary approach.
We hope this addition clarifies the point raised.
- Please introduce the flowchart leading to the selected studies.
We thank you for this comment. According to the Editors’ requests, we have clarified that our paper has not been conceived as a Systematic Review. Thus, we maintained the search strategy to guarantee the transparency of our work, but we did not add a flowchart, that was not requested for a typical Review article.
- Were markers of ketosis measured in the selected studies? This would enable to compare the efficacy of the diets in all the selected studies.
Thank you for your insightful comment. In response, we have added, study by study, details on how ketosis was measured in the selected studies.
- The paragraph on adverse effects: it is not clear whether the authors have extracted the information from the selected studies... the data from the selected studies could be more cross-referenced.
Thank you for your insightful comment. In response to your suggestion, we have now revised the paragraph on adverse effects to ensure that the information is clearly extracted from the selected studies. Additionally, we have cross-referenced the data from the selected studies to enhance clarity and provide a more robust evidence base for the reported adverse effects.
- The authors' conclusions are more the sum of the individual findings of the various articles than a complete synthesis of these different studies.
Thank you for your valuable feedback. In response to your comment, we have revised the discussion section to provide a more comprehensive summary of the studies, rather than merely listing the individual findings. This revision aims to offer a more cohesive synthesis of the results from the selected studies, addressing the broader implications and patterns that emerge from the data.
- Regarding gender disparities, several papers have been published on the subject by Mauvais-Jarvis. This author should be cited.
Thank you for pointing out the relevant studies by Mauvais-Jarvis. In response to your comment, we have added a reference to his studies in lines 293-298, providing a brief discussion of his contributions to the topic.
We hope that our responses and the revisions we have made to the manuscript have addressed your valuable comments and suggestions. We have made every effort to ensure that the revised version reflects the necessary improvements, and we trust that the updates will meet your expectations.
Thank you once again for your insightful feedback and for your time and consideration in reviewing our work.
Round 2
Reviewer 2 Report
Comments and Suggestions for Authors
The comments have been processed correctly, with the exception of this one: ""It is not mentioned whether lowering body weight in the same way as following a ketogenic diet would have a higher benefit? a lower benefit? with regard to hepatic steatosis."
new comment: The authors must correctly answer the question posed above.
Indeed, the authors responded: "Beyond its well-established effect on body weight, the ketogenic diet has demonstrated 157 significant benefits in improving insulin resistance. Notably, these effects on insulin re-158 sistance are, at least in part, independent of weight loss, thereby expanding the potential 159 applications of this dietary approach" which is not the response of my question
Authors must correctly answer the question
Author Response
Comment 1: The comments have been processed correctly, with the exception of this one: ""It is not mentioned whether lowering body weight in the same way as following a ketogenic diet would have a higher benefit? a lower benefit? with regard to hepatic steatosis."
new comment: The authors must correctly answer the question posed above.
Indeed, the authors responded: "Beyond its well-established effect on body weight, the ketogenic diet has demonstrated significant benefits in improving insulin resistance. Notably, these effects on insulin re-158 sistance are, at least in part, independent of weight loss, thereby expanding the potential applications of this dietary approach" which is not the response of my question.
Authors must correctly answer the question.
Author's R1:
Dear Reviewer, Thank you for your valuable suggestions and comments on our manuscript.
We have addressed your comment by citing the EASL-EASD-EASO clinical practice guidelines ((https://doi.org/10.1016/j.jhep.2024.04.031).
Line 179-189 : "These effects highlight the potential therapeutic role of the ketogenic diet in patients with hepatic steatosis, even in the absence of weight loss, and support its efficacy in this clinical setting.
However, the EASL-EASD-EASO clinical practice guidelines state that the role of the keto-genic diet in the prevention and treatment of MASLD is not supported by significant evidence, although it does not exclude a positive impact on liver health. Currently, there are no spe-cific recommendations for a ketogenic diet approach to managing MASLD, unlike a Med-iterranean diet approach and physical activity. Considering that in adults with MASLD and overweight, weight loss induced by dietary and behavioral therapy could improve hepatic steatosis (with at least a 5% weight reduction), we can assume that the ketogenic diet could help to achieve this goal in this patient category "